# ASSEMBLAGE: Automatic Binary Dataset Construction for Machine Learning

**Chang Liu**[1]*, **Rebecca Saul**[2]*, **Yihao Sun**[1], **Edward Raff**[2,3],
**Maya Fuchs**[2], **Townsend Southard Pantano**[1], **James Holt**[4], **Kristopher Micinski**[1]
[1]Syracuse University, [2]Booz Allen Hamilton,
[3] University of Maryland, Baltimore County, [4]Laboratory for Physical Sciences
cliu57@syr.edu, Saul_Rebecca@bah.com, ysun@syr.edu, Raff_Edward@bah.com,
fuchs_maya@bah.com, tgsoutha@syr.edu, holt@lps.umd.edu, kkmicins@syr.edu

## Abstract

Binary code is pervasive, and binary analysis is a key task in reverse engineering, malware classification, and vulnerability discovery. Unfortunately, while there exist large corpora of malicious binaries, obtaining high-quality corpora of benign binaries for modern systems has proven challenging (e.g., due to licensing issues). Consequently, machine learning based pipelines for binary analysis utilize either costly commercial corpora (e.g., VirusTotal) or open-source binaries (e.g., coreutils) available in limited quantities. To address these issues, we present ASSEMBLAGE: an extensible distributed system that crawls, configures, and builds Windows PE binaries to obtain high-quality binary corpora suitable for training state-of-the-art models in binary analysis. We have run ASSEMBLAGE on AWS over the past year, producing 890k Windows PE and 428k Linux ELF binaries across 29 configurations. ASSEMBLAGE is designed to be both reproducible and extensible, enabling users to publish "recipes" for their datasets, and facilitating the extraction of a wide array of features. We evaluated ASSEMBLAGE by using its data to train modern learning-based pipelines for compiler provenance and binary function similarity. Our results illustrate the practical need for robust corpora of high-quality Windows PE binaries in training modern learning-based binary analyses. ASSEMBLAGE code is open sourced under the MIT license, and the dataset can be downloaded from https://assemblage-dataset.net/.

## 1  Introduction

Binary analysis, the task of understanding and analyzing an executable binary program, is extraordinarily labor-intensive. Practitioners often spend hours to weeks manually analyzing a single program [57]. The development of automated binary analysis tools has also proven to be extremely labor-intensive. Historically, such tools have relied on careful and judicious software engineering efforts to handle all corner cases, or used SAT solvers and similar data structures to resolve as many conditions as possible [45]. Given this high human effort, it is unsurprising that an increasing number of binary analysis methods use machine learning to help automate such efforts. This includes low-level tasks such as function disassembly [70], and high-level tasks like malware detection [59] or variable name identification [14, 39] which leverage these lower-level results.

However, significant issues in data availability have hampered the development of machine learning-based binary analysis for decades [1, 56]. Disassembly and decompilation require a mapping between the original true source or assembly code and the corresponding bytes of binary instructions, but

---

*Equal contributions

38th Conference on Neural Information Processing Systems (NeurIPS 2024) Track on Datasets and Benchmarks.

it is non-trivial to collect and maintain such a mapping. Malware detection requires access to a collection of "benign" programs to train on, but can not be legally distributed due to copyright and license concerns. The situation is particularly bad for Windows Portable Executable (PE) binaries: while Windows is the most frequent target of malware, modern binary analysis systems train almost exclusively on Linux ELF binaries. Toolchain identification requires knowledge of the entire compilation process and of a variety of compiler settings that are simply not recorded in the original file. Binary function similarity requires having a ground truth of functions that are semantically equivalent but written or compiled differently; most existing models rely on coding competition data that does not reflect the diversity of real-world code. The ad-hoc approaches taken by prior works to tackle these issues further complicate the reproducibility of these methods due to the infrastructure challenges in setting up an environment to build the same tools and perform complex data pre-processing [25, 47, 54, 55]. Moreover, authors often do not have the necessary licenses or do not make the code publicly available for others to extend the work.

We introduce ASSEMBLAGE, a tool that makes significant progress towards overcoming these challenges. ASSEMBLAGE is a framework for crawling code hosting platforms, downloading open-source projects, compiling them with multiple different compiler versions and settings, and recording the necessary information to reconstruct byte-level mappings from the output back to the source code. By using open-source code targets from GitHub, we obtain greater diversity in programs, resolve licensing/distribution issues, and provide detailed ground truth to support all the aforementioned research areas. This alleviates the concerns with non-semantic preserving modifications [22] and the exorbitantly expensive cost and more limited diversity of binary re-writing [11, 68]. ASSEMBLAGE is designed to be extensible so that additional sources of code, compilers, and feature extractors can be added for further functionality. ASSEMBLAGE datasets can be further distributed as "recipes," a list of repositories and settings so that issues with mixed-license repositories (e.g., GPL and BSD) do not make a binary-based distribution unlicensable. Finally, we have designed and tested the system to ensure the reproducibility of datasets built by ASSEMBLAGE.

Our contributions include:

**(1)** ASSEMBLAGE, a cloud-based distributed system carefully engineered to automatically construct diverse corpora of high-quality Windows PE binaries while ensuring compliance with licensing and terms of service.

**(2)** The ASSEMBLAGE dataset, consisting of Windows and Linux binaries with a total of 29 different combinations of CPU architecture, optimization level, compiler, compiler version, and other compiler flags.

**(3)** Three case studies that demonstrate ASSEMBLAGE's application to (a) compiler provenance detection, (b) function similarity identification and (c) function boundary identification. Our case studies demonstrate the need for ASSEMBLAGE's rich features, particularly for corpuses of Windows PE binaries.

## 2   Related Work

**The PE Binary Availability Crisis**   Some prior works have looked at crawling open-source repositories to build datasets [10, 23, 43, 61] and leverage various build systems (such as DIRTY, XDA, and DIRE) [38]. Unfortunately, all these prior works focus on Linux binaries and build systems due to their ease of use. We found that nearly half of the transformer-based models rely on executables from standard Linux packages such as *coreutils*, *binutils*, *diffutils*, *findutils*, *busybox*, *libcurl* and *openssl*. With only one recent transformer-based model even testing on Windows PE Wu et al. [69]. The heavy Linux reliance [15, 28, 31, 62] and use of ubiquitous libraries [3, 17, 30, 36, 47] is problematic since most malware is Windows PE, and these utilities do not reflect the breadth of code types in the wild. Work that does focus on Windows data is almost exclusively distributed in a pre-processed and featurized form, forgoing the original binaries due to licensing issues [4, 27, 41, 53, 60, 72, 73], or purely malware and lacks any source-binary mappings [34, 58].

The lack of datasets with available source-to-binary mapping is also a major impediment to current research trends in the binary analysis space. Lessons/models from Natural Language Processing in developing BERT [19] and GPT [50] have shown that both architecture and data scaling are key factors to improved results [12, 67]. While many works are building base binary models that can be

Table 1: Comparison between ASSEMBLAGE and related datasets. Prior datasets are often restricted in availability, do not have source-binary mappings, and are limited in platforms/toolchains, making it difficult to train and test for the diversity of real-world applications for binary analysis. Column names are abbreviated as; N/A: not available. L: Linux. W: Windows. G: GCC. C: Clang. M: MSVC. MD: malware detection. CP: compiler provenance. FS: function similarity. FN: function name/type recovery. DC: decompilation. PTY: The dataset is proprietary and for commercial license use. U: undisclosed or unknown.

| Dataset | Binaries (#) | Available data format | Functions (#, K) | Projects (#) | OS | | Toolchain | | | Supporting task | | | | |
|---|---|---|---|---|---|---|---|---|---|---|---|---|---|---|
| | | | | | L | W | G | C | M | MD | CP | FS | FN | DC |
| SPEC CPU2006 [5, 49] | 981 | PTY | U | 7 | ✓ | ✓ | ✓ | ✓ | ✓ | | | | | ✓ |
| Ubuntu Dataset[6] | 87,853 | Binaries | 88,000 | 22,040 | ✓ | | ✓ | ✓ | | | | | ✓ | |
| PF[52] | 2,132 | N/A | 1,362 | 5 | ✓ | | ✓ | ✓ | | | | | ✓ | |
| PSI[52] | 188, 253 | N/A | U | 14,000 | ✓ | | U | U | | | | | ✓ | |
| Nero[18] | 542 | Binaries | U | U | ✓ | | U | U | U | | | | ✓ | |
| DIRE[39] | 164,632 | Binaries | 3,196 | U | ✓ | | U | U | | | ✓ | | ✓ | |
| BinKit[36] | 243,128 | Binaries | 75,231 | 51 | ✓ | | ✓ | ✓ | | | | ✓ | | |
| Passtell[21] | 552 | Binaries | 150 | 15 | ✓ | | ✓ | ✓ | | | ✓ | | | |
| DIRT[13] | 75,656 | Binaries | 998 | U | ✓ | | ✓ | | | | ✓ | | ✓ | |
| BinaryCorp -26M[66] | 48,130 | Binaries | 25,877 | 9,819 | ✓ | | ✓ | ✓ | | | ✓ | ✓ | | |
| BinBench[16] | 1,127,479 | Binaries | 4,408 | U | ✓ | | ✓ | ✓ | | | ✓ | ✓ | ✓ | |
| LLM4- Decompile[65] | U | Binaries | U | 164 | ✓ | | ✓ | | | | | | | ✓ |
| **Assemblage** | **1,536,171** | **Binaries +PDB** | **783,694** | **220,792** | ✓ | ✓ | ✓ | ✓ | ✓ | ✓ | ✓ | ✓ | ✓ | ✓ |

fine-tuned to perform a wide variety of tasks [7, 37, 41, 74], they predominantly use datasets that are too small or not diverse, as shown in Table 1 with a median of 3.7k binaries per LLM based model. Wang et al. [66] and Zhu et al. [74] are trained with the BinaryCorp dataset, which, at 48,130 (Linux) binaries, the largest prior work that still lacks Windows binaries, is not sufficiently diverse in binary behavior (as our results will show), and short of the 890k Windows PE and 428k Linux ELF binaries ASSEMBLAGE provides. This is also true of works that tackle specific downstream tasks, including binary code similarity detection [2, 24, 44, 66], compiler provenance identification [29], vulnerability detection [71], and malware detection [26, 40, 69].

**Modern Efforts in Binary Analysis Ignore Windows**  With the increasing use of binary analysis in security-relevant settings, the absence of publicly available corpora of benign Windows binaries is a critical concern. The difficulty of obtaining Windows PE data in academic literature is prevalent in related domains like malware detection that rely on small datasets [34], effect sizes [51], and label noise [32, 33, 35], which generally depend on VirusShare and VirusTotal for data. An examination of files uploaded to VirusShare in 2023 indicates that 40.38% of the roughly 1.8 million malware samples are Windows binaries, whereas less than one percent are ELF binaries [58].[2] Given the prevalence of Windows executables, and the fact that Windows accounts for roughly seventy percent of global operating system market share [64], it is troubling that nearly all the recent literature on machine learning approaches to binary analysis uses Linux binaries as training data. This is especially worrying because, as we demonstrate in Section 4, models trained on Linux binaries do not necessarily generalize to the Windows domain. Therefore, in order to produce useful machine learning models for binary analysis, it is necessary at a minimum to test, and likely necessary to train, these models on Windows data.

---

[2]The other major file types are PDF, which accounts for roughly 25% of files, and Unknown, which accounts for 32% of files and captures files whose types could not be parsed. Files in the unknown category include data files, some scripts, ASC files, and source code.

# 3 The ASSEMBLAGE Dataset

## 3.1 Dataset Generation Pipeline

The overview of ASSEMBLAGE is illustrated in Figure 1 and Figure 2. The grey area indicated in Figure 1 shows ASSEMBLAGE's workflow, and it is implemented as a distributed system running on multiple servers for scalability. We also release the codes along with the dataset, and provides development and deployment documentation for it, more details are enclosed in our supplemental materials. First, the crawler begins by either sending queries to GitHub API to discover new repositories or crawl existing webpage hosting projects to obtain source codes. Then, source codes are cloned and modified to assign compiler flags such as optimization level and CPU architecture. On Windows, ASSEMBLAGE uses the Microsoft Visual Studio tool chain, incorporating a set of automation scripts to overcome Visual Studio's reliance on GUI-based user interfaces; while on Linux, workers are implemented for both GCC and Clang, using Docker to facilitate flexible and isolated deployment. After compiling is completed, all files will be collected to extract information (e.g., function address, source codes, comments) and stored in files. The last step is to compress all files for storage.

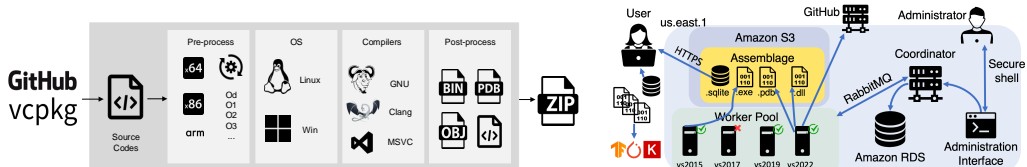

Figure 1: ASSEMBLAGE's build pipeline. Grey boxes are run in a parallel, distributed manner.

Figure 2: ASSEMBLAGE architecture.

## 3.2 Dataset Details and Distribution

**890k Windows-GitHub PE Dataset**  We constructed a dataset of 890K Windows PE binaries as follows: starting from 4.6 million crawled GitHub repositories dating from 2008 to 2023, we discarded repositories that had fewer than 10 files, were smaller than 50KB in size, lacked a README.md file containing the string "Visual Studio", or lacked a solution file supporting Microsoft Visual Studio tool chains. We were left with 179,548 repositories, from which ASSEMBLAGE was then able to build and store binaries using our Microsoft Visual Studio tool chains. Build configurations varied in toolchain version (Visual Studio 2015, 2017, 2019, and 2022), optimization level (0d, 01, 02, 0x), CPU architecture, and Debug/Release mode. In sum, we ended up with 847,856 executable (.exe) files and 43,507 dynamically-linked library (.dll) files, along with PDB files for each binary. In total, we accumulated 2TB worth of data, of which the executables (.exe and .dll files) took 200GB. To conform with ethical data distribution practices, we recorded license information from crawled repositories when available.

We measured our dataset along a variety of dimensions to demonstrate its quality and diversity. Figure 3 shows a histogram of sizes of binaries from our (a) Windows binaries from GitHub, (b) vcpkg and (c) Linux binaries from GitHub corpora.

Our Windows-GitHub binaries were the smallest overall, with an average size of 143KB. Manual inspection revealed many student projects, basic utilities, and starter codes in our GitHub dataset. However, in absolute terms, our Windows-GitHub dataset still contains a substantial number of large binaries, including 26,549 binaries $\geq$ 0.5MB. Note that compiling random Windows projects is non-trivial due to a lack of standardization; more details are provided in Appendix B.

We also inspected the function information corresponding to our Windows-GitHub PE dataset. We found that out of 252M functions, there are 14M unique function names, and 138M unique function hashes. ASSEMBLAGE was also able to store source code for 17M functions in its SQL database. We also observed significant variation in the number of functions per binary, which we illustrate in Figure 4. Finally, we examined the relationship between a binary's size and the functions it contains. Counter-intuitively, the number of functions in a binary is not correlated with the size of that binary, as these variables have a Pearson correlation coefficient of 0.22. Furthermore, the size of functions within a binary has a 0.49 correlation coefficient to the binary size, but no correlation to the number of

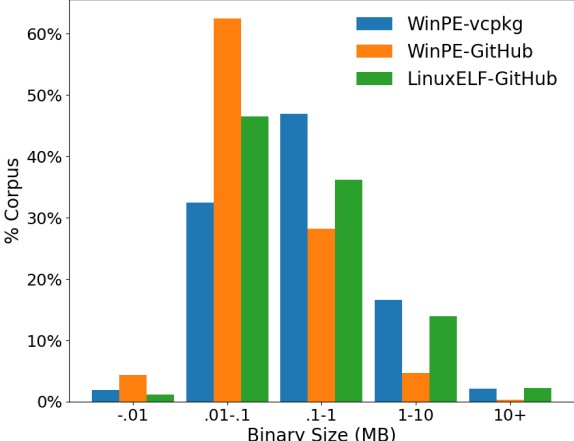

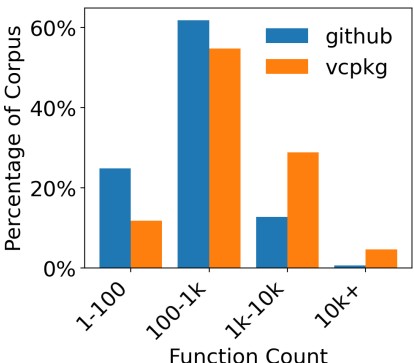

Figure 3: Binary size, Windows PE (GitHub), vcpkg, and Linux ELF (GitHub).

Figure 4: Number of functions in each binary.

functions in the binary. We unexpectedly conclude that binary size doesn't have a strong correlation with either the number of functions nor the average function size in a binary. Moreover, the existence of function-level information provides insights into the behavior patterns of specific compilers. For example, in 2017 Visual Studio implemented stack-based buffer protection by inserting security cookies, and we can locate more than 744k `__security_init_cookie` functions with 151k unique hashes. These function level information provided by ASSEMBLAGE presents valuable data for not only machine learning research but also a potential application to a wide range of research areas like security and binary analysis.

**25k Windows vcpkg Dataset** Upon examining our Windows-GitHub dataset, we observed that only 4.7% of the corpus consisted of library files. To address this deficiency and provide coverage of Windows' statically-linked and dynamically-linked libraries, we adapted ASSEMBLAGE to build binaries from `vcpkg` [48], a dataset of 2,185 actively-developed, well-maintained, and widely-used packages. `vcpkg` consists of many commonly used and permissively licensed C++ libraries, including graphics and UI-related libraries.

Our `vcpkg` worker takes advantage of the native commands of `vcpkg`, and we modified the package manager's original build script to pass custom variables and flags to its compiler. In this way, we harvested a diverse Windows PE library corpus from `vcpkg`, comprising 24905 binaries, along with their pdb files, from 2078 different projects. Only 110 of these projects are also present in our Windows-GitHub dataset, demonstrating the added diversity that is present in our `vcpkg` dataset. Similarly to the Windows-GitHub dataset, `vcpkg` binaries were built using 4 different versions of the Microsoft Visual Studio toolchain, as well as with 4 optimization levels (`-O{d,1,2,3}`).

Finally, as illustrated in Fig. 3, the size of the binaries in our `vcpkg` dataset is larger than the size of those in our Windows-GitHub dataset. The `vcpkg` binaries were 2124 KB on average, compared to 143KB for Windows-Github binaries, with a small number (roughly 500) of `vcpkg` binaries $\geq$ 10 MB.

**Proof-of-Concept Linux Dataset** Motivated by the scarcity of corpora of benign Windows PE binaries, we have focused our efforts on building such executables. However, to ensure ASSEMBLAGE's API was sufficiently expressive, we also added several Linux builders. Our prototype Linux crawler traverses GitHub and searches for `Makefiles` to identify potential Linux ELF projects. Among 4 million crawled repositories, ASSEMBLAGE recognized 367,207 repositories that include a `Makefile`. We created 8 build configurations, using GCC (`-O{0,1,2,Z}`) and Clang (`-O{0,1,2,3}`).

Similar to what we did when building our Windows PE database, we deployed ASSEMBLAGE to AWS with the same coordinator, but with workers on different EC2 instances. In sum, ASSEMBLAGE generated 428,884 Linux binaries from 30,856 GitHub repositories. The histogram of the file size distribution of our Linux ELF dataset is shown in Figure 3, the composition of compiler and

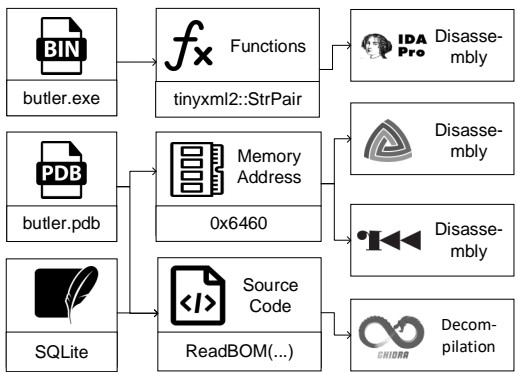

Figure 5: Dataset database and usage

optimization level of the dataset is presented in Table 5, and the overall dataset size is recorded in Table 1. Interestingly, we observe a much higher frequency of licensed repositories in our Linux-GitHub dataset; nearly half of our Linux-GitHub binaries have licenses, compared to only 11% of our Windows-GitHub binaries.

Generally, our Linux builder failed more often than our Windows PE builder (a success rate of 7% compared to the 30% on Windows), but produced larger files in the projects that it did build. Though it is natural to be concerned about build failures during the dataset creation process, these errors are not introduced by ASSEMBLAGE, as we simply use regular expression to substitute the compiler flags stored in the Make files, then call make to start the building. Indeed, a manual inspection confirmed that most of the build errors resulted from errors in the repository's source code or Make file.

In addition to the GitHub crawler, we also built a proof-of-concept crawler and builder that leverages the Arch User Repository (AUR) for source code. Users can set the configuration by providing custom makepkg.conf for each project. While the AUR repositories are well maintained by the community and open-sourced, due to the rate limits and ethical concerns, we did not deploy builders to generate a large scale dataset from AUR. However, our proof-of-concept crawler demonstrates that ASSEMBLAGE can be configured to work with a variety of package managers.

**ASSEMBLAGE Dataset Distribution Format**  We only release the binaries built from repositories that clearly state their license, and a typical ASSEMBLAGE dataset consists of both (a) an SQLite database of metadata and (b) an archive of binaries (either as a `.tar` or manually via the Amazon S3 interface). The SQL database provides a logical specification of a collection of binary files, currently `.exes` and `.dlls`. The user queries the SQLite database to perform application-specific feature extraction (e.g., identifying pairs of sibling functions for triplet learning), and often feeds the binaries into a machine learning framework such as TensorFlow, PyTorch, or Keras.

The SQLite database consists of four primary tables: `binaries`, `functions`, `rvas`, and `lines` (illustrated Figure 5). The binaries table retains a reference of each binary's configuration used during compilation, e.g. platform, build mode, compiler version, and optimization, and assigns each binary a unique identifier for further reference. We also register additional information about each executable including the file name, the location of the project on GitHub, the project's license, and the size of the binary in this table. Using the binaries table, researchers can curate subsets of the dataset to meet their training needs, e.g. by selecting only binaries of a certain size or with a certain license.

To provide function-level details and minimize the need to distribute PDB files (since, as we saw earlier, PDB files can take up 90+% of dataset space), the `functions`, `rvas` and `lines` tables reference the primary key for each binary stored in `binaries` table to relate functions and binaries. The `functions` tables offers a high-level overview of functions, presenting a hash of the function's bytes and the function name, whereas the `rvas` and `lines` tables provide lower-level function details. Specifically, the `rvas` table documents the relation between each function and its relative virtual address when the binary is loaded into memory. On the other hand, the `lines` table illustrates the mapping of each source code line to the precise byte address located in the compiled binary. The dataset also provides the relation between binaries and their corresponding PDB files in a `pdbs` table, in case researchers want to make use of those files for further binary analysis.

Figure 5 shows the information that is available about a single binary within a ASSEMBLAGE dataset. As an example, we query the database for a binary called `butler.exe`, which has the unique identifier `184456`.

In the binaries table, we get basic facts about the executable, including the compiler version (Visual Studio 2017), build mode (Release), optimization flag (O2), and associated license (MIT). In the functions table, we can see that this binary has 568 functions; in particular, we focus on one function called `tinyxml2::StrPair::SetStr`, uniquely identified by the tag `51629995`. We use this unique identifier to query the `lines` and `rvas` tables for further information about this function. In the `lines` table, we can see the source code corresponding to this function, such as the line `if (*(pu + 0) == TIXML_UTF_LEAD_0`. From the `rvas` table, we learn that this function will be loaded into the relative addresses from `0x6460` to `0x64AA`. Thus, this example demonstrates how all the information needed to do binary analysis with corpuses of ASSEMBLAGE executables is available in the accompanying ASSEMBLAGE SQLite database.

**Ethical Concerns** It is natural to be concerned about the presence of malware in any corpus of binary executables. ASSEMBLAGE builds sources from GitHub, which prohibits malware and allows users to report suspected malware to its community discussion board. Currently, ASSEMBLAGE does not independently check for malware; we are looking to add malware scanning, following the model pioneered by EMBER [4], in the next iteration of the tool. In addition, the public GitHub repository links are included in our metadata (we do not provide author-level metadata, but plan to distribute this in the future in normalized form), which links the binary data to the author's GitHub profile. Our public datasets include only those repositories for which permissive licenses may be obtained; we facilitate archiving corpora of unlicensed repositories by distributing "recipes."

## 4 Benchmarks

In this section, we demonstrate the utility of ASSEMBLAGE for a variety of machine learning tasks. Three widely-regarded machine learning approaches to binary analysis problems are evaluated under their original protocols, baring the minimum possible changes to support/read Windows PE files over the original Linux files: a LightGBM model for compiler provenance identification [21], a graph neural network (GNN) for binary function similarity detection [42], and a BERT-based transformer network for understanding binary files, fine-tuned on the task of binary function similarity detection [66]. These experiments broadly show that the strong results of prior literature do not yet generalize to Windows and more diverse compilation settings, and that ASSEMBLAGE provides the data to further explore these topics (amongst others).

Experiments were run on an Ubuntu 22.04 server equipped with an AMD EPYC 7713P@1.9Ghz 64-Core Processor and 512GB of RAM.

### 4.1 Compiler Provenance

Compiler provenance inference is the task of determining the compiler configuration used to build a binary, including compiler version, build flags, and compiler passes. We trained a state-of-the-art provenance learning model PassTell [21] on a combination of data, mixing the authors' original Linux dataset with Windows PE binaries obtained from ASSEMBLAGE. We first used ASSEMBLAGE to build a dataset of Windows binaries that complemented the Linux binaries studied in the original PassTell paper. We attempted to build each of the 15 source projects selected by Du et al. [21], but due to differences between the Linux and Windows platforms and their respective compiler infrastructures, some projects could not be rebuilt. In total, we collected 127 binaries, which we proceeded to preprocess via the same protocol as the original paper. Though the PassTell authors used `objdump`, a linear disassembler, to extract function bytes, we found higher success rates with IDA Pro, augmented with the PDB files collected by ASSEMBLAGE, to accurately handle the significant number (5%) of discontiguous functions present in Windows binaries. Following disassembly, we masked operands and literals in the same manner as Du et al. [21], and balanced the training data by compiler toolchain, version, and optimization level.

We evaluate PassTell twice, once using the aforementioned mix of original Linux and new Windows PEs and a second time using only Windows PEs. The confusion matrix in Figure 6 shows that the PassTell approach over-fits the original Linux models (top left diagonal), but has greater difficulty

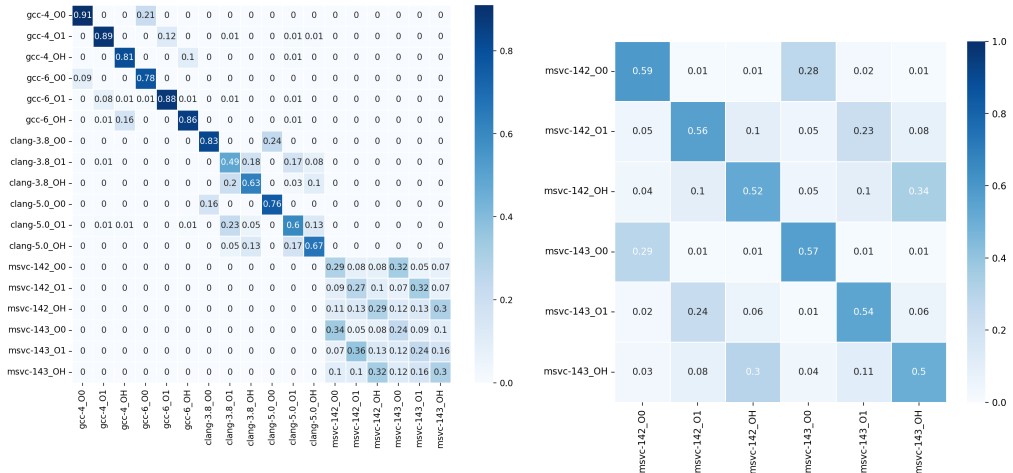

Figure 6: PassTell's Confusion Matrix on the dataset mixing the author's training data (top left) with ASSEMBLAGE Windows PE binaries (bottom right).

Figure 7: PassTell's Confusion Matrix on the ASSEMBLAGE's Windows PE Binaries.

Table 2: AUC scores evaluating binary function similarity model generalization between the Linux and Windows domains. Column names indicate train/test platform.

| Task | Windows/ Linux | Linux/ Windows | Linux/ Linux | Windows/ Windows |
|---|---|---|---|---|
| arch | 0.50 | — | 0.97 | — |
| bit | 0.70 | — | 0.99 | — |
| comp | 0.63 | — | 0.80 | — |
| opt | 0.72 | 0.62 | 0.88 | 0.89 |
| ver | 0.82 | 0.64 | 0.98 | 0.84 |
| XA | 0.48 | — | 0.86 | — |
| XC | 0.63 | — | 0.86 | — |
| XC+XB | 0.61 | — | 0.87 | — |
| XM | 0.54 | 0.61 | 0.87 | 0.86 |

with the Windows PE files (bottom right). To determine if this is a capacity issue, we re-evaluate on only the Windows PEs to reduce the number of classes and make the ML problem easier, but Figure 7 shows that PassTell still struggles on Windows PEs with $\approx 58\%$ accuracy. The observation of PassTell's performance on Windows data indicates the limits of training models on small binary datasets lacking diversity. While the approach put forth by Du et al. [21] seemed promising when applied to a small Linux dataset, extending the work to Windows data from ASSEMBLAGE revealed serious shortcomings. This underscores the utility of ASSEMBLAGE as a tool for machine learning researchers in the binary analysis space.

## 4.2 Binary Function Similarity

Next, we used ASSEMBLAGE to evaluate how well a graph neural network (GNN) trained on Linux data generalizes to Windows binaries.

We conducted these experiments on the GNN developed by Li et al. [42] for binary function similarity detection, as implemented and open-sourced by Marcelli et al. [47].[3]

---

[3]Code from https://github.com/Cisco-Talos/binary_function_similarity/.

Table 3: Reproducing jTrans on ASSEMBLAGE Windows and Linux datasets, measuring in MRR (higher is better).

| Training Epochs | 1 | 20 | 50 |
|---|---|---|---|
| Fine-tune & Evaluate on Windows | 0.17 | 0.52 | 0.52 |
| Fine-tune & Evaluate on Linux | 0.83 | 0.83 | 0.83 |
| Fine-tune & Evaluate on Linux (BinaryCorp-26M) | 0.82 | - | 0.98 |

Table 4: F-1 scores of XDA and IDA on Windows PE binary function boundary identification

| Dataset | XDA | IDA | IDA (w/PDBs) |
|---|---|---|---|
| WinPE | 0.75 | 0.47 | 0.79 |
| vcpkg | 0.81 | 0.86 | 0.86 |

We made only one change to their data processing pipeline — instead of removing singleton functions, we allowed them to stay in the dataset and be part of negative pairs, as they can still provide a training signal to the model this way and improved performance in a small-scale test.

Using ASSEMBLAGE and Marcelli et al. [47]'s protocol, we end up with 1.78 million functions for triplet learning. Negative pairs consist of two different functions, whereas positive pairs consist of functions originating from the same source code, but differing in one or more of the following: architecture, bitness, compiler, compiler version, and optimization. Many tasks/pairs that make sense in Linux (e.g., ARM vs. PowerPC and endianness changes) are not widely applicable in Windows PE code bases, and so can not be evaluated in this context. For this reason, Table 2 shows Linux evaluation results on the left side, the original results in Linux/Linux column, and Windows evaluation results on the right for the viable tests. Irrespective, we see that despite the cross-platform/functional goals of Marcelli et al. [47], the dichotomy in Windows PEs and more diverse datasets results in dramatically lower performance. The GNN model trained on ASSEMBLAGE Windows data shows that, despite using the same code and sophisticated disassembly tools, the gap in performance is non-trivial and an open problem for future study.

A second experiment using the BERT [20] based jTrans [66] is considered. jTrans uses masked language modeling and a novel task, jump target prediction, in its pre-training phase. Despite the jump target prediction code released by Wang et al. [66] requiring Linux specific tooling, all information needed to run their fine-tuning was already in ASSEMBLAGE's SQLite database. Otherwise, we precisely followed the script released by the jTrans authors to create a training and validation set.

We randomly selected 72,000 Windows PE binaries and 23,000 Linux ELF binaries from ASSEMBLAGE's dataset, processed them, and separated them into training and validation sets for Windows and Linux, respectively. For both the Windows and Linux datasets, using the hyperparameter settings recommended by the jTrans authors, we ran fine-tuning for 50 epochs, at which point the Mean Reciprocal Rank (MRR) over the training set became stable. Table 3 shows a notable difference between the reported MRR scores on Linux (0.83) and Windows (0.52) data, even with fine-tuning. Because the authors have not released their code for pre-training, we could not conduct any further investigation into the impact of the pre-training step. However, it is significant that jTrans and jTrans-zero (jTrans without fine-tuning) both perform worse on the larger, more diverse ASSEMBLAGE data than they do on their original training corpora. This demonstrates the value that ASSEMBLAGE brings to researchers, who can leverage its datasets to conduct experiments on more challenging and more realistic data.

## 4.3 Binary Function Identification

Finding functions within a binary is non-trivial due to the complexity of inverting compiler behaviors, as well as the general difficulty of disassembly as its own requisite task [5, 8, 63]. This is often assumed to be solvable with $\geq 99\%$ $F_1$ scores for ML based function boundary detection [53] or that professional reverse engineering tools like IDA are sufficiently accurate [5, 9, 18, 46, 52, 53, 66]. ASSEMBLAGE allows us to quantify the accuracy of these tools on a more diverse collection of data

Windows data[4]. We randomly selected around 2000 Windows PE binaries from ASSEMBLAGE that better reflect "wild" code, and 2500 `vcpkg` binaries that are closer to the limited Windows PE binaries often considered in prior works. As seen in Table 4, significant room for improvement still exists for ML-based methods like XDA [53], and do provide real value over existing rules-based methods like IDA. Even when IDA is given debug information (PDB files), its performance is not sufficient to discount in results, analysis, and real-world deployment.

## 5 Conclusion

At first blush, binary analysis fits into a common application pattern of deep learning: an inversion problem where the compiler's process of converting source code into binary files is the easy generating pass, and machine learning methods are used to learn the inversion from binary back to source code. ML research in binary analysis has thus exploded, but has often not translated to practice due to a gap between "in-the-lab" code and "in-the-wild" binaries. ASSEMBLAGE helps to mitigate this gap and provides an extensible system that can generate the data needed to push this needed research into the Windows domain, which has the greatest need and the least data/study. This has been shown for three subdomains of binary analysis of compiler provenance, function similarity, and function identification. In each case, we see ASSEMBLAGE makes it easy to set up experiments following existing methods, obtain new results, identify areas for improvement, and ultimately stable ASSEMBLAGE's general purpose utility for binary analysis research.

## Acknowledgments

We would like to acknowledge the feedback of the anonymous NeurIPS dataset track reviewers. Additional computing infrastructure (GPUs and servers) was provided by Syracuse University.

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

## A    Technical documentation and dataset access

The homepage of ASSEMBLAGE is `https://assemblage-dataset.net/`, documentation for ASSEMBLAGE can be accessed at the link `https://assemblagedocs.readthedocs.io`, and the binary dataset built from repository with license is hosted on Hugging Face and accessible by everyone, all the links are listed in the documentation.

We also release the codes we used to build the dataset on GitHub, link at `https://github.com/Assemblage-Dataset/Assemblage`. Anyone can obtain the copy of our source code and replicate our dataset with our recipes. We also provide specific instructions on deployment.

## B    Note on the difficulty of compiling random projects

We also present a breakdown of our Windows-GitHub dataset by compiler and optimization level (Table 5). The number of binaries for each compiler version and optimization is not equal due to unevenly distributed computing power among ASSEMBLAGE worker nodes, changes to the availability of repositories, restricted running time, and the limited availability of some configurations in solution files.

Table 5: The distribution of configurations (opt: optimization level). The number of files at each optimization level is realatively even, but outliers exist for a variety of reasons. Projects do not compile effectively under different optimization settings without manual intervention, cause time-out issues in taking too long to compile, or a variety of other factors. How to automatically build more projects is an open question for future work.

(a) WinPE-Github

| Compiler | Opt | Count |
| --- | --- | --- |
| MSVC-v140 | O1 | 64,598 |
| | O2 | 177,531 |
| | Ox | 107,635 |
| MSVC-v141 | O1 | 6,503 |
| | O2 | 84,237 |
| | Od | 93,155 |
| | Ox | 7,547 |
| MSVC-v142 | O1 | 91,501 |
| | O2 | 53,062 |
| | Od | 76,139 |
| MSVC-v143 | O1 | 85,171 |
| | Od | 59,582 |
| | Ox | 40,004 |

(b) WinPE-vcpkg

| Compiler | Opt | Count |
| --- | --- | --- |
| MSVC-v141 | O1 | 2,864 |
| | O2 | 2,864 |
| | Od | 2,624 |
| | Ox | 2,855 |
| MSVC-v142 | O1 | 8,053 |
| | O2 | 8,077 |
| | Od | 6,230 |
| | Ox | 7,888 |
| MSVC-v143 | O1 | 3548 |
| | O2 | 3548 |
| | Od | 3,519 |
| | Ox | 3,282 |

(c) LinuxELF-Github

| Compiler | Opt | Count |
| --- | --- | --- |
| GCC-11.4.0 | O0 | 66,919 |
| | O1 | 56,919 |
| | O3 | 61,907 |
| | Oz | 27,952 |
| Clang-14.0.0 | O0 | 74,947 |
| | O1 | 75,973 |
| | O2 | 81,093 |
| | O3 | 83,963 |

To further investigate the varying number of binaries under each build configuration, we recorded the error message encountered during build failures, which we summarize in Table 6. The most common causes of build failure are related to compiler infrastructure, such as invalid solution configurations, Windows SDK errors due to upgrading, invalid or missing project files, and other compilation errors. Although we tried to mitigate the errors caused by missing files by installing a wide range of SDK libraries, complete coverage was infeasible, and thus, certain library versions were absent from our worker environment. We also did our best to minimize invalid configuration errors by minimally modifying variables in vcxproj files and using `devenv` from MSVC to upgrade solution files when appropriate.

Moreover, we manually inspected the log files associated with certain build configurations that had particularly low success rates: `v141-O1-Debug-x64`, `v141-Ox-Debug-x64`, and `v140-O1-Release-x86`. We present the results in Table 7. We found that 86% of `v141-O1-Debug-x64` and 66% of `v141-Ox-Debug-x64` failures produce error code MSB4126,

Table 6: Most common error codes from ASSEMBLAGE Windows dataset

| Error Code | Ratio | Memo |
|---|---|---|
| MSB4126 | 47% | Invalid specified solution configuration |
| C1083 | 18% | Missing source file |
| MSB8036 | 8% | Specified Windows SDK version was not found |
| MSB3202 | 3% | Missing project file |
| MSB4025 | 3% | The imported project file loading error |

which indicates that the solution configuration is invalid. By comparison, the dominant error code for `v140-O1-Release-x86` is C1083, which indicates that a certain file is missing or wrong.

Table 7: Error codes for specific build configuration

| | Flags | MSB4126 | C1083 | MSB8036 | MSB4025 | Others |
|---|---|---|---|---|---|---|
| MSVC-v140 | O1-Release-x86 | 25% | 34% | 0% | 1% | 40% |
| | O2-Debug-x64 | 82% | 5% | 0% | 0% | 13% |
| | O2-Release-x86 | 25% | 34% | 0% | 1% | 40% |
| | Ox-Release-x64 | 21% | 36% | 0% | 0% | 43% |
| MSVC-v141 | O1-Debug-x64 | 86% | 3% | 0% | 4% | 6% |
| | O2-Release-x86 | 30% | 24% | 0% | 6% | 40% |
| | Od-Release-x86 | 37% | 21% | 0% | 6% | 35% |
| | Ox-Debug-x64 | 66% | 10% | 0% | 6% | 18% |
| MSVC-v142 | O1-Release-x64 | 20% | 23% | 30% | 1% | 26% |
| | O2-Release-x86 | 25% | 22% | 28% | 1% | 25% |
| | Od-Debug-x64 | 69% | 7% | 12% | 0% | 12% |
| MSVC-v143 | O1-Debug-x86 | 25% | 18% | 33% | 6% | 18% |
| | Od-Debug-x64 | 84% | 4% | 2% | 4% | 6% |
| | Ox-Release-x86 | 18% | 36% | 16% | 2% | 28% |

## C   Privacy concerns on open source projects

There exists raising concerns on the privacy issue on open source project hosting platforms. Git commits consists of username, email and GitHub's repository URL scheme naturally embeds user and the project name. This information can be used to keep track of code changes and facilitate collaboration in large open source projects. Due to the raising amount of tools and work [13, 38, 39] that rely on open source projects from GitHub to build dataset, the privacy of open source contributors has come to more researchers' attention.

ASSEMBLAGE doesn't index the repository by its owner, but the owner's username is revealed in the GitHub URL, which is a necessary reference to the original source code. In addition, we will not mirror the original GitHub repositories for public consumption so that when owners change a repository's visibility or turn on anonymous email in GitHub settings, they have effective control of data inclusion in future recipes.

