# OpenReview forum: "Assemblage: Automatic Binary Dataset Construction for Machine Learning"
_NeurIPS.cc/2024/Datasets_and_Benchmarks_Track — NeurIPS 2024 Track Datasets and Benchmarks Poster_

### Official Review · Reviewer_RpLW · 2024-07-23
**Review for ASSEMBLAGE: Automatic Binary Dataset Construction for Machine Learning**

**Rating:** 8
**Confidence:** 4

**Review:**

**Quality:**
This work is of high quality, offering a comprehensive and well-structured dataset tailored for machine learning-based binary code analysis. The authors have meticulously designed ASSEMBLAGE to be reproducible and extensible, ensuring it meets the diverse needs of researchers. The extensive dataset, coupled with detailed metadata, provides a robust foundation for various binary analysis tasks.

**Clarity:**
The paper is clearly written, with thorough explanations of the ASSEMBLAGE framework, the dataset generation process, and the evaluation through case studies. Each section is well-structured, making it easy to follow the authors' methodology and reasoning. The inclusion of figures and tables to illustrate the dataset's characteristics and the framework's architecture further aids in understanding the work's scope and significance.

**Originality:**
ASSEMBLAGE is original, addressing the critical challenge of obtaining high-quality benign binaries for machine learning applications. While there are similar datasets such as SPEC[2]; and Ubuntu[3], they are too old and may not position themselves as datasets for ML-based binary analysis. Meanwhile, ML-based datasets like AnghaBench[1] do not specifically focus on facilitating learning-based approaches. None of these match the scale and diversity offered by ASSEMBLAGE. The dataset's extensive configurations, including various compiler versions and optimization levels, set it apart from existing datasets, providing a richer resource for binary code analysis.

**Significance:**
The significance of this work is substantial, as it provides a crucial resource for advancing binary code analysis. By generating a comprehensive dataset of Windows PE and Linux ELF binaries, ASSEMBLAGE fills a major gap in the availability of benign binaries for machine learning models. The dataset's public availability and detailed documentation promote transparency and encourage further research. The data collection process also serves as a valuable reference for future projects, setting a new standard for dataset creation in this field.


[1] Da Silva A F, Kind B C, de Souza Magalhães J W, et al. Anghabench: A suite with one million compilable c benchmarks for code-size reduction[C]//2021 IEEE/ACM International Symposium on Code Generation and Optimization (CGO). IEEE, 2021: 378-390.

[2] Nair A A, John L K. Simulation points for SPEC CPU 2006[C]//2008 IEEE International Conference on Computer Design. IEEE, 2008: 397-403.

[3] Shariati M, Dehghantanha A, Martini B, et al. Ubuntu One investigation: Detecting evidences on client machines[J]. arXiv preprint arXiv:1807.10448, 2018.

**Strengths:**

1. **Well-motivated dataset:** The paper clearly articulates the motivation and importance of creating a comprehensive binary dataset. It emphasizes the scarcity of high-quality benign binary datasets and sets a strong foundation by explaining the critical need for ASSEMBLAGE in advancing machine learning-based binary analysis. This detailed context underscores the necessity and relevance of the work, making a compelling case for its contribution to the field.

2. **Clear data generation process:** The paper provides a thorough and well-structured explanation of the dataset generation process, detailing each step from crawling repositories to configuring and building binaries. The inclusion of visual aids, such as figures illustrating the pipeline and architecture, enhances comprehension. This meticulous description ensures that other researchers can replicate the process, highlighting the framework's reproducibility and robustness.

3. **Diversity of the collected data:** The paper showcases the comprehensive and diverse nature of the dataset, emphasizing its unique contributions compared to other existing datasets like those from GitHub[1] and vcpkg[2]. By detailing the extensive configurations, including various compiler versions and optimization levels, it demonstrates the dataset's richness and versatility. This diversity makes ASSEMBLAGE a valuable resource for a wide range of binary analysis tasks, setting it apart from other datasets.

4. **Practicality in collection and application:** The paper demonstrates the practical applicability and significance of the dataset through detailed benchmarks and case studies. By presenting evaluation results for tasks such as compiler provenance detection and binary function similarity, it underscores the dataset's effectiveness in improving the performance and reliability of binary analysis models. These benchmarks provide concrete evidence of the dataset's value to the research community, highlighting its potential to drive advancements in the field.

[1] Shao H, Sun D, Wu J, et al. paper2repo: Github repository recommendation for academic papers[C]//Proceedings of The Web Conference 2020. 2020: 629-639.

[2] Interrante-Grant A, Wang M, Baer L, et al. Synthetic Datasets for Program Similarity Research[J]. arXiv preprint arXiv:2405.03478, 2024.

**Additional Feedback:**

No additional feedback. The paper is well-structured and addresses the key aspects of dataset creation and evaluation effectively. The detailed documentation and comprehensive approach provide a solid foundation for further research and application.

**Clarity:**

The paper is clearly written, with thorough explanations of the ASSEMBLAGE framework, the dataset generation process, and the evaluation through case studies. Each section is well-structured, making it easy to follow the authors' methodology and reasoning. The inclusion of figures and tables to illustrate the dataset's characteristics and the framework's architecture further aids in understanding the work's scope and significance.

**Correctness:**

**Correctness:**

1. **Claims Validity:** The claims made in the submission appear to be correct, with the authors providing a thorough justification for the necessity and utility of ASSEMBLAGE. The dataset is constructed in a sound manner, utilizing a comprehensive pipeline to ensure reproducibility and extensibility.

2. **Dataset Construction:** The construction of the dataset follows a robust process, involving the crawling, configuring, and building of binaries from diverse open-source projects. The use of multiple compiler versions and optimization levels adds to the dataset's reliability and diversity.

3. **Evaluation Methods:** The evaluation methods and experiment design are appropriate and performed correctly. The authors use relevant benchmarks and case studies to demonstrate the dataset's utility in various machine learning-based binary analysis tasks, providing clear evidence of ASSEMBLAGE's effectiveness.

**Documentation:**

Yes, the paper provides sufficient detail on data collection and organization, explaining the dataset generation process and the configurations used. The dataset is available at the provided URL (https://assemblage-dataset.net/), with clear information on hosting, licensing, and maintenance. The authors address ethical and responsible use by discussing licensing considerations and potential future enhancements. Overall, the documentation supports reproducibility and responsible use of the dataset.

**Ethics:**

No, I do not suspect there are any ethical concerns with the submission that warrant further discussion or review. The dataset appears to comply with data privacy, copyright, and consent guidelines, and there are no apparent issues related to data quality, representativeness, safety, security, discrimination, bias, fairness, deception, harassment, environmental impact, or human rights.

**Limitations:**

**Limitations:**

1. **Addressing Dataset Limitations:** The authors could more thoroughly address the limitations of the ASSEMBLAGE dataset. For example, they could discuss any biases introduced by the specific selection of open-source projects and compilers used. Additionally, they could elaborate on any constraints related to the diversity and representativeness of the binaries included in the dataset.

2. **Potential Negative Societal Impact:** While ASSEMBLAGE aims to advance binary analysis for benign purposes, the authors should acknowledge and address the potential for misuse. They could discuss the ethical considerations and safeguards that need to be in place to prevent the dataset from being used for malicious purposes, such as creating or enhancing malware.

3. **Future Work and Mitigation Strategies:** The authors should outline potential future work to address the identified limitations and mitigate any negative societal impacts. For instance, they could suggest methods for further diversifying the dataset, improving the robustness of the binaries, and implementing ethical guidelines for the responsible use of ASSEMBLAGE. By being upfront about these aspects, the authors can set a positive precedent for transparency and responsibility in research.

**Constructive Suggestions for Improvement:**

- **Expand on Dataset Limitations:** Provide a more detailed discussion on the biases and constraints of the ASSEMBLAGE dataset, including the selection process for open-source projects and compilers.
- **Ethical Considerations:** Address the potential negative societal impact of the dataset, particularly the risk of misuse for malicious purposes. Discuss safeguards and ethical guidelines for the responsible use of ASSEMBLAGE.
- **Future Directions:** Suggest concrete future work and mitigation strategies to enhance the dataset's diversity, robustness, and ethical use. This could include proposing additional sources of binaries, improving dataset generation methods, and establishing best practices for ethical research.

**Opportunities For Improvement:**

**Comparison with existing datasets:** While the motivation for creating ASSEMBLAGE is clearly articulated, the paper could benefit from a more detailed discussion of existing datasets and their limitations. Providing a comprehensive comparison with similar datasets would further underscore the unique contributions of ASSEMBLAGE and highlight its distinct advantages.

**Technical details:** The description of the dataset generation pipeline, while thorough, could include more technical details about the challenges encountered and how they were addressed. For example, elaborating on the specific issues with compiler flags and build configurations, and how the team overcame them, would provide valuable insights for other researchers looking to replicate or extend this work.

**quantitative analysis of data diversity:** The paper could enhance its discussion of the dataset's diversity by including more quantitative analysis. For instance, providing statistical summaries or visualizations of the distribution of binaries across different configurations, compiler versions, and optimization levels would give a clearer picture of the dataset's richness and variability.

**Limitation and future work:** While the benchmarks and case studies demonstrate the utility of ASSEMBLAGE, the paper could expand on the limitations and potential areas for future work. Discussing any observed shortcomings in the dataset or the evaluation methods, and proposing ways to address these in future iterations, would add depth to the analysis and suggest a clear path for ongoing research.

**Relation To Prior Work:**

Yes, the paper clearly discusses how ASSEMBLAGE differs from previous contributions. It highlights the limitations of existing datasets such as AnghaBench, SPEC, and Ubuntu, emphasizing that they either lack the scale, diversity, or specific focus required for machine learning-based binary analysis. ASSEMBLAGE is positioned as more comprehensive and versatile, offering extensive configurations and a larger dataset that better supports a wide range of binary analysis tasks.

**Summary And Contributions:**

In this paper, the authors propose a binary dataset named ASSEMBLAGE, specifically designed to enrich benign binaries by generating over 890k Windows PE and 428k Linux ELF binaries from open-source projects. They claim that ASSEMBLAGE is reproducible and extensible, ensuring diverse configurations across various compiler versions and optimization levels. The dataset and code are publicly available here (https://assemblage-dataset.net/, promoting transparency and further research.

---

> ### Author Rebuttal · Authors · 2024-08-15
>
> > Limitation and future work...
>
> The largest limitation of our work today is that it relies on projects being sufficiently "plug-and-play" that they can be compiled in an automated fashion. This is not true for many large real-world projects on the Windows platform, where specific install instructions must be followed. A second-order version of this limitation is that non-standard compilers, such as the Intel compilers, will require additional effort to integrate into Assemblage's build process. These issues motivate our ongoing work with Assemblage, which include both new compilers and platforms, along with new infrastructure for building large projects (such as Chromium) which necessitate custom build infrastructure. We carefully engineered Assemblage's extensible architecture to facilitate these goals.
>
>  >Potential Negative Societal Impact ... Future Work and Mitigation Strategies
>
> Regarding ethical considerations, we have taken care to ensure we are compiling open-source software, which is a requirement for hosting it on GitHub. There is a potential issue in users who have not understand what it means to open-source their software and the nature of the rights they are extending to others, which we will create an informational page and document per our response to reviewer XvnZ. This will include the information needed for authors to take steps to change their repository if they realize they do not wish to be potentially included in a dataset. We will not mirror the original GitHub repos for public consumption so that when they change a repository's visibility, they have effective control of that data's inclusion in future recipes.
>
> > the paper could benefit from a more detailed discussion of existing datasets and their limitations.
>
> Table 1 was meant to serve this purpose, but we will add additional exposition to the table. In particular, prior dataset releases do not provide the infrastructure for the dataset's construction itself. All prior datasets we are aware of take the low-resistant path of using Linux or Linux-adjacent (e.g., MinGW) toolchains to create their datasets due to better standardization and package management. Still, most are small (except for BinBench) in the number of binaries and, thus, the total number of functions available. This also means they lack Windows binaries that reflect the majority of real-world needs. Because Assemblage is designed around a framework rather than a singular release, we have made it easier to support various tasks like malware detection, compiler provenance, function similarity, function name/type recovery, and decompilation research. Per Table 1, most datasets are made to support only a limited set of tasks, with no readily available means to extend them to add new tasks. Since Assemblage provides the recipes and APIs for extension, future tasks can be added as desired.
>
> >Technical details: The description of the dataset generation pipeline, while thorough, could include more technical details about the challenges encountered and how they were addressed.
>
> One surprisingly major issue we observed that--beyond GitHub’s publicly-stated rate limits--GitHub also imposed secondary limits that resulted in significant throttling of API accesses on a per-IP basis after several hours. Thus, we designed Assemblage to employ exponential backoff to ensure we respected both the publicly-stated and both secondary rate limits we discovered throughout the course of mining repositories.
>
> We will document other insights in the revised version, this is a thoughtful and valuable exercise to help capture the often "hidden" work of making research progress.
>
> >quantitative analysis of data diversity: The paper could enhance its discussion of the dataset's diversity by including more quantitative analysis.
>
> Please see the attached PDF for a new figure we will add that breaks up the number of files built by operating system, compiler, architecture, and optimization level. One can see that "simply" building programs with different compilers and settings is highly non-trivial, with varying success rates across settings and versions. It is the nature of such problems that we can not expect even distributions in all cases, and real-world data reflects this wide asperity (e.g., people don't really deploy `O0` optimized code, but it is technically an option in Linux).
>
> We hope this has helped address your questions and concerns. Thank you for your valuable review and time, please let us know if there is anything further we can clarify.

---

> > ### Comment · Reviewer_RpLW · 2024-08-15
> >
> > Thank you for your detailed rebuttal. I am raising my score to 8 to further improve the chances of this work being accepted. Great work!

---

### Official Review · Reviewer_XvnZ · 2024-07-27
**Useful binary dataset**

**Rating:** 7
**Confidence:** 4
**Correctness:** There are no soundness constraints.
**Clarity:** The paper is well-written and easy to…

**Review:**

Assemblage can crawl, configure, and build Windows PE binaries, producing high-quality binary corpuses. Over the past year, Assemblage has generated 890k Windows PE and 428k Linux ELF binaries on AWS. It is designed for reproducibility and extensibility, allowing users to publish dataset "recipes" and extract various features. Assemblage's data has been used to train pipelines for compiler provenance and binary function similarity, highlighting the need for robust corpuses of high-quality Windows PE binaries in modern binary analysis.

**Strengths:**

* Cloud-Based Distributed System: It is a robust, cloud-based system designed to automatically create diverse, high-quality corpora of Windows PE binaries, while adhering to licensing and terms of service.

* Comprehensive Dataset: It includes Windows and Linux binaries across 29 different combinations of CPU architecture, optimization levels, compiler types, compiler versions, and other compiler flags.

* Diverse Applications: It has been applied in case studies for compiler provenance detection and function similarity identification, showcasing its versatility and the necessity of its rich features.

* Licensing Compliance: Takes into account compliance with licensing requirements, making it a reliable tool for generating and distributing datasets without legal issues.

* Extensibility: Designed to be extensible, allowing for the addition of new code sources, compilers, and feature extractors, thus enhancing its functionality over time.

* Reproducibility: Ensures the reproducibility of datasets, providing a reliable foundation for future research and development in binary analysis.

**Additional Feedback:**

No other additional feedback.

**Documentation:**

https://assemblage-dataset.net/ has good documentation of the dataset. The code is available at https://github.com/Assemblage-Dataset/Assemblage

**Ethics:**

There are no ethics concerns.

**Limitations:**

It would be good to add discussion on how such a dataset could be possibly misused for malware development or violating privacy of developers.

**Opportunities For Improvement:**

The code documentation at https://github.com/Assemblage-Dataset/Assemblage can be improved.

**Relation To Prior Work:**

Data availability issues have hindered the development of machine learning-based binary analysis for decades. The challenges include creating accurate mappings between source code and binary instructions, acquiring benign programs for malware detection, and dealing with licensing issues, especially for Windows PE binaries. These problems have led to a reliance on Linux ELF binaries and complicated reproducibility due to varied infrastructure setups and the lack of publicly available code. ASSEMBLAGE addresses these challenges by providing a framework for crawling code hosting platforms, downloading open-source projects, compiling them with various compiler versions and settings, and recording detailed information for byte-level mappings back to the source code. Using diverse open-source targets from GitHub, ASSEMBLAGE resolves licensing issues and supplies ground truth for various research areas. It is extensible, allowing additional code sources, compilers, and feature extractors, and supports distributing datasets as "recipes" to avoid licensing conflicts.

**Summary And Contributions:**

Binary code is essential for reverse engineering, malware classification, and vulnerability discovery, but obtaining high-quality corpuses of benign binaries is difficult due to licensing issues. Current machine learning pipelines for binary analysis rely on costly commercial corpuses or limited open-source binaries. To address this, Assemblage, an extensible cloud-based distributed system, is proposed.

---

> ### Author Rebuttal · Authors · 2024-08-15
>
> > The code documentation ... can be improved.
>
> We agree with the importance of having better documentation. We are in the process of more complete documentation, which we plan to host using ReadTheDocs. Our current progress is available at https://assemblagedocs.readthedocs.io/en/latest/index.html and already a significant improvement over the previous documentation.
>
> > It would be good to add discussion on how such a dataset could be possibly misused for malware development or violating privacy of developers.
>
> Regarding the need for a discussion on how our dataset could be misused to violate the privacy of developers, we appreciate that this is an issue. We note that Assemblage currently does not provide more information than is available publicly via Git histories; the dataset contains no private information beyond what may already be found on GitHub. Similarly, we do not host code which is unlicensed in our public dataset (instead preferring to index these repositories via recipes), to avoid running afoul of the developers' intentions. Similarly, because our dataset contains a snapshot of repositories at a certain point in time, it lags GitHub deletions. It is certainly possible that a project author may have open-sourced their code without realizing the ramifications of what the OSS license allows and would retroactively not have made it OS because they do not wish to be included in a project like Assemblage. We will add a discussion of these issues to the paper.
>
> We will add instructions on how users can remove their projects from GitHub or make them private, and since we do not store/replicate repositories, this effectively allows the user to remove themselves from all future iterations of Assemblage and any further attempts to download their code. We will also add recommendations for how an author may upload their code to GitHub in a more privacy-preserving way (i.e., scripts to change email and names) so that they may also balance making their code open source while preserving more anonymity.
>
> Thank you for your time and review. Please let us know any other questions we can clarify on our work.

---

> > ### Comment · Reviewer_XvnZ · 2024-08-15
> > **Thank!**
> >
> > Thank you for the clarifications.

---

### Official Review · Reviewer_X8os · 2024-07-29
**Large Windows binrary software datasets for machine learning**

**Rating:** 6
**Confidence:** 3
**Correctness:** Yes.
**Clarity:** Yes.

**Review:**

Analyzing Windows binary files is an important topic for various applications. The provided dataset, ASSEMBLAGE, ended up with 847,856 executable (.exe) files and 43,507 dynamically-linked library (.dll) files. Experiments on Compiler Provenance, Binary Function Similarity, and Binary Function Identification show the limitation of existing machine learning algorithms, especially for these methods mainly focusing on Linux. Such dataset provides potential for future machine learning algorithm development for Windows binary analysis.

**Strengths:**

1. Windows binary analysis is an important topic.
2. The dataset is large with diverse categories of Windows files.
3. Benchmark results indicate the limitations of existing ML algorithms.

**Additional Feedback:**

Minor Typos:

1. Line 259: “recipes.”
2. Line 270: RAM..

**Documentation:**

Yes.

**Limitations:**

Yes.

**Opportunities For Improvement:**

1. The dataset does not classify the malware, which limits the applicability for Malware Detection. It can potential provide some showcase for reverse engineering, malware classification, and vulnerability discovery.

2. In table 2, what is the performance on "Window/Windows"?

**Relation To Prior Work:**

Yes.

**Summary And Contributions:**

Analyzing binary code is the key task for reverse engineering, malware classification, and vulnerability discovery. Most former datasets focus on Linux system and the windows dataset has limited quality. By clawing on github, the paper provides a high-quality Windows dataset with 890k Windows PE. Benchmark experiments on machine learning tasks show the practical need for robust corpora of high-quality Windows PE datasets.

---

> ### Author Rebuttal · Authors · 2024-08-15
>
> We sincerely appreciate your taking the time to review the paper.
>
> > The dataset does not classify the malware...
>
> You are correct that we are not releasing malware as a part of this dataset. This is intentional, as many resources currently exist for obtaining malware samples. VirusShare is a free resource that is continuously updated. The SOREL-20M dataset has over 10 million malware samples available for download, and a variety of other tools exist, such as "honey pots" to collect malware "in the wild." For this reason, we argue that there is no gap in malware availability, and we instead focus on tackling the gap in benign binaries (particularly Windows binaries).
>
> We note that the lack of malware does not impose much limitation on vulnerability discovery, as the source code for malware is not available. Current research in vulnerability discovery will predominantly use (just a few) well-known libraries and instrument them or develop new bespoke detection processes to find (and then verify) vulnerabilities, which Assemblage enables at a larger scale via automation.
>
> >In table 2, what is the performance on "Window/Windows"?
>
> In terms of the Windows/Windows performance, we attempted to create the training/testing set in the same manner as the paper [A], two issues prevent all tasks from being possible:
>
> 1. The tasks comp, XC, XC+XB require more than one compiler, but most all Windows C/C++ projects are designed to compile only with the Microsoft Visual C/C++ compiler. While we have different versions of the compiler, that is not sufficient for this task.
>
> 2. The task XA and XA+XO requires the binaries to have different CPU architecture, though some of our binaries are cross-compiled into ARM64 architecture, x86 is the dominant CPU architecture for Windows, and almost all projects we crawled are written for x86.
>
> For these reasons, the tasks that can be meaningfully completed today in a Windows/Windows setting is *XM*, *ver* and *opt*, where *XM* received a score of 0.86, *ver* received a score of 0.84 and *opt* received a score of 0.89 in our testing. We will add the above context to the paper. Our hope is that Assemblage's extensibility as a system will allow others to do research, like converting Microsoft Visual Studio project files to compile with non-standard Windows compilers, so that more diversity can be generated. The ability of Assemblage to enable these new research areas  and directions is an important part of our contribution.
>
> A: Andrea Marcelli, Mariano Graziano, Xabier Ugarte-Pedrero, Yanick Fratantonio, Mohamad Mansouri, and Davide Balzarotti. How machine
>
> We hope this has addressed your questions, please let us know anything we can clarify further.

---

> > ### Comment · Reviewer_X8os · 2024-08-28
> >
> > Thank you for the detailed response and I will keep my rating.

---

### Comment · Area_Chair_G94q · 2024-08-27
**Reminder to respond to author rebuttals**

Dear Reviewers,

Please remember to respond to the authors' rebuttals before the end of the discussion period on the 31st of August as they have put in a lot of effort to respond to your concerns! If you have already responded, thank you.

Regards,
Your AC

---

### Decision · Program_Chairs · 2024-09-26

**Decision:**

Accept (Poster)

**Comment:**

This paper presents a well-documented and comprehensive dataset of Windows binaries that can be used to train machine learning models for tasks such as malware classification and vulnerability discovery. All the reviewers agreed that the paper is well-motivated, provides a comprehensive dataset and is both significant and original.